# *Saccharomyces cerevisiae* goes through distinct metabolic phases during its replicative lifespan

Simeon Leupold[1†], Georg Hubmann[1†‡], Athanasios Litsios[1], Anne C Meinema[1§], Vakil Takhaveev[1], Alexandros Papagiannakis[1#], Bastian Niebel[1], Georges Janssens[2¶], David Siegel[3], Matthias Heinemann[1]*

[1]Molecular Systems Biology, Groningen Biomolecular Sciences and Biotechnology Institute, University of Groningen, Groningen, Netherlands; [2]European Research Institute for the Biology of Ageing, University of Groningen, University Medical Centre Groningen, Groningen, Netherlands; [3]Analytical Biochemistry, Groningen Research Institute of Pharmacy, University of Groningen, Groningen, Netherlands

*For correspondence:
m.heinemann@rug.nl

[†]These authors contributed equally to this work

Present address: [‡]Laboratory of Molecular Cell Biology, Department of Biology, Institute of Botany and Microbiology, Center for Microbiology, KU Leuven, VIB, Heverlee, Belgium; [§]Institute of Biochemistry, ETH Zurich, Zürich, Switzerland; [#]Microbial Sciences Institute, Yale University, West Haven, United States; [¶]Laboratory Genetic Metabolic Diseases, Amsterdam Gastroenterology and Metabolism, Amsterdam UMC, University of Amsterdam, Amsterdam, Netherlands

Competing interests: The authors declare that no competing interests exist.

**Abstract** A comprehensive description of the phenotypic changes during cellular aging is key towards unraveling its causal forces. Previously, we mapped age-related changes in the proteome and transcriptome (Janssens et al., 2015). Here, employing the same experimental procedure and model-based inference, we generate a comprehensive account of metabolic changes during the replicative life of *Saccharomyces cerevisiae*. With age, we found decreasing metabolite levels, decreasing growth and substrate uptake rates accompanied by a switch from aerobic fermentation to respiration, with glycerol and acetate production. The identified metabolic fluxes revealed an increase in redox cofactor turnover, likely to combat increased production of reactive oxygen species. The metabolic changes are possibly a result of the age-associated decrease in surface area per cell volume. With metabolism being an important factor of the cellular phenotype, this work complements our recent mapping of the transcriptomic and proteomic changes towards a holistic description of the cellular phenotype during aging.
DOI: https://doi.org/10.7554/eLife.41046.001

## Introduction

Cellular aging is a complex multifactorial process affected by an intertwined network of effectors such as protein translation, protein quality control, mitochondrial dysfunction and metabolism (*Barzilai et al., 2012*; *Kennedy et al., 1994*; *Lagouge and Larsson, 2013*; *Webb and Brunet, 2014*). Disentangling cause and effect is a major challenge in aging research (*McCormick and Kennedy, 2012*). A key requisite towards unraveling the causal forces of cellular aging is a comprehensive account of the concomitant phenotypic changes. In the replicatively aging budding yeast *Saccharomyces cerevisiae*, a common model for mitotic aging (*Eisenberg et al., 2007*), unfortunately, the application of cell ensemble-based omics methods has been difficult due to the rapid outgrowth of aging mother cells by the newly formed daughter cells. Through a novel cultivation technique, allowing us to generate large amounts of aged cells, we could recently perform proteome and transcriptome profiling throughout the whole lifespan of *S. cerevisiae*. There, on the basis of an identified gradually increasing uncoupling between protein and transcript levels of biogenesis-related genes, we conjectured that this uncoupling is one of the causal forces of aging (*Janssens et al., 2015*). Furthermore, we found changes in expression of enzymes and, consistent with an earlier report (*Lin et al., 2001*), in metabolic genes, suggesting an altered metabolism with increasing replicative age. Here, exploiting our novel cultivation technique (recently also adopted by

others; *Hendrickson et al., 2018*), metabolomics and model-based inference methods (*Niebel et al., 2019*), we identified a metabolic shift during the replicative lifespan of *S. cerevisiae*. With this work, we complement our recent proteome and transcriptome profiling data with the corresponding metabolome and fluxome, and generate a description of the functional phenotypic changes accompanied with cellular aging which ultimately lead to senescence and cell cycle arrest.

## Results

### Column-based cultivation to enrich aged mother cells

To generate large quantities of aged cells, required for the metabolic profiling, we used our earlier developed column-based cultivation technique. Here, biotinylated cells attached to streptavidin-conjugated iron beads are immobilized inside a column positioned in the center of a ring magnet. A continuous nutrient flow through the column removes emerging daughter cells, while largely retaining mother cells (*Janssens et al., 2015*). Several columns operated in parallel, allowed harvesting cells at different time points, corresponding to cell age. In order to be able to infer data for aged cells from the harvested samples (which still contained a fraction of daughter cells), we generated at each harvesting time point three samples differently enriched with aged mother cells; (1) from the column effluent, (2) from the column after an additional washing step, and (3) from the washing solution (in the following referred to as mix 1, 2 and 3) (*Figure 1*). The exact sample compositions (i.e. the fraction of mother, daughter and dead cells) were determined by flow cytometry using a combined dye-staining with propidium iodide and avidin–FITC. We then determined the cell population-averaged intracellular metabolite concentrations and, to assess physiological parameters, measured the change in extracellular metabolites concentrations due to cell growth over a period of 3 hr. To infer the aged mother cells' metabolite levels, physiological parameters and intracellular metabolic fluxes from the mixed-sample measurements, we employed different mathematical model-based methods (*Figure 1*).

### Intracellular metabolite concentrations decrease with cell age

The intracellular concentrations of 18 metabolites, mainly located in central carbon metabolism, were quantified by LC-MS/MS in the differently mixed samples (i.e. mix 1, 2 and 3), taken at various time points (after 10, 20, 44 and 68 hr). As these concentration measurements resembled the average concentration of metabolites originating from mother and daughter cells, we used non-negative linear regression to infer the metabolite concentration in each individual population (i.e. aged mother and young daughter cells), using the determined fractional abundances of each population and the age-dependent cell volumes, which we determined with microfluidics and microscopy (*Figure 2—figure supplement 1*). To confirm the validity of the regression approach, where in general a good fitting was achieved ($R^2$ = 0.89) (*Figure 2—figure supplement 2*), we compared the concentrations for daughter cells, inferred from the mixed population samples, with metabolite concentrations independently determined from a culture of young streptavidin-labeled cells. Here, we found an excellent agreement between mathematically inferred and directly determined concentrations ($R^2$ = 0.99) (*Figure 2—figure supplement 3*).

Focusing on the intracellular metabolite dynamics in aging mother cells, we found that the concentrations of all quantified metabolites already at a relatively young age start to decrease on average to about half of their initial values (*Figure 2a* and *Figure 2—figure supplement 4*). Previously, also other phenotypic changes have been observed at a young age (*Janssens and Veenhoff, 2016*). Remarkably, despite the drop in ATP levels, the adenylate energy charge was maintained between 0.8 and 0.95 (*Figure 2—figure supplement 5*), which corresponds to values of exponentially growing cultures (*Ditzelmüller et al., 1983*). The drop in metabolic concentrations suggests that metabolic activities are globally decreased in aged cells and, as many metabolites have also regulatory function (*Huberts et al., 2012*; *Litsios et al., 2018*), the observed concentration changes are expected to lead to metabolic rearrangements.

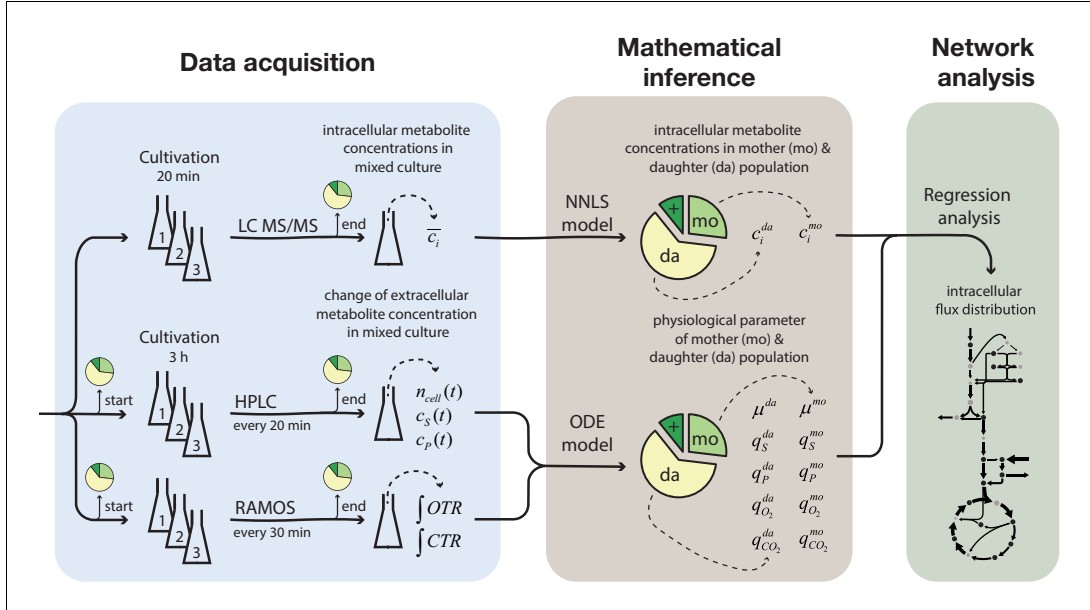

**Figure 1.** Overview of the experimental and model-based analyses to determine the metabolite levels, physiological parameters and intracellular metabolic fluxes of replicatively aging budding yeast. Samples were harvested at various time points (corresponding to different cell ages) from a column-based cultivation system (*Janssens et al., 2015*), designed to enrich aged mother cells. The fractional abundance of mother, daughter and dead cells in each sample was determined by flow cytometry and a combined dye-staining with propidium iodide and avidin–FITC. Aliquots were used to determine the intracellular metabolite concentrations, $\bar{c}_i$, by LC-MS/MS and the cell count, $n_{cell}(t)$, by flow cytometry, extracellular metabolites (i.e. substrates and products), $c_S(t)$ and $c_P(t)$, by HPLC and the integral of oxygen and carbon transfer rates, *OTR* and *CTR* (i.e. total consumed oxygen and produced carbon dioxide) by a Respiration Activity Monitoring System (RAMOS), in the mixed population samples. Next, the age-dependent intracellular metabolite concentrations ($c_i$) were inferred from the acquired population-average data using non-negative least square regression (NNLS) and the physiological parameters (growth ($\mu$) and metabolite exchange rates ($q$)) of mother (mo) and daughter (da) cells) from an ordinary differential equation (ODE) model. The inferred physiological parameters and intracellular metabolite levels of aged mother cells were then analyzed using a combined stoichiometric-thermodynamic metabolic model and regression analysis to obtain the intracellular metabolic flux distribution.
DOI: https://doi.org/10.7554/eLife.41046.002

## Cells switch from a fermentative to a respiratory metabolism with age

To assess changes on the level of metabolic fluxes, we next determined the physiological rates, that is growth, metabolite uptake and excretion rates of aging cells. At each time point (after 10, 20, 44 and 68 hr), we measured the evolution of cell count and extracellular concentrations of glucose, pyruvate, acetate, glycerol and ethanol over a period of three hours in each harvested sample (i.e. mix 1, 2 and 3). The fractional abundance of each cell population was determined before and after that period. We used a second set of aliquots to measure the evolution of produced carbon dioxide and consumed oxygen using a Respiration Activity Monitoring System (RAMOS) (*Hansen et al., 2012*). To infer the population-specific physiological rates from the mixed-population samples, we fitted the acquired dynamic data to an ordinary differential equation model, describing the changes of the biomass and extracellular metabolite concentrations in the samples, due to mother and daughter cell growth and their respective metabolism (*Figure 2—figure supplements 6–8*). To assess the validity of the inference approach, we compared the physiological rates inferred for daughter cells to physiological rates independently determined from unlabeled as well as from streptavidin-labeled cell cultures, both consisting of predominantly young cells. Here, we found a good agreement between the rates mathematically inferred for daughter cells and the rates directly obtained from these cultures containing young cells (*Figure 2b*).

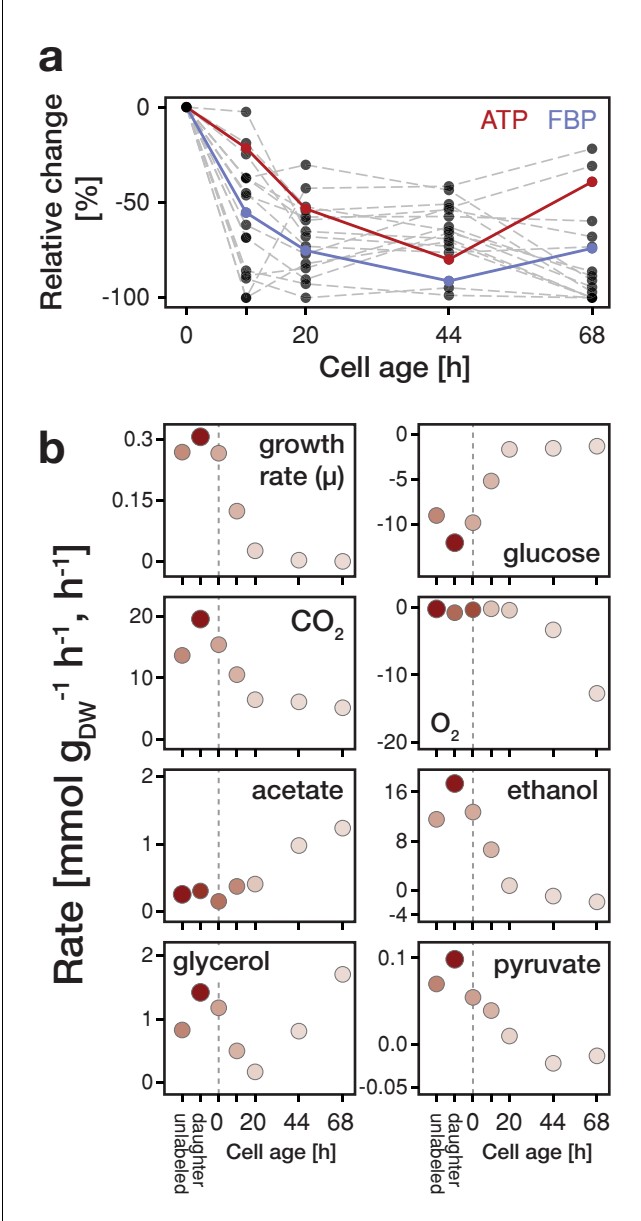

**Figure 2.** Changes in metabolite concentrations and physiological parameters during cellular aging. (a) The intracellular metabolite concentrations of 18 metabolites at various cell ages were inferred from LC-MS/MS measurements, cell volume measurements and the fractional abundances of each cell population using non-linear least square regression. Grey dashed lines depict the change of intracellular metabolite concentrations relative to concentrations determined from streptavidin-labeled cells (i.e. young cells at an age of 0 hr). The change in ATP concentration is highlighted in red, and FBP (fructose-1,6-bisphosphate) in blue. *Figure 2—figure supplement 4* shows the data for each metabolite in absolute units. *Figure 2—source data 1* contains the data. (b) The growth (μ), metabolite uptake and production rates at various cell ages were obtained by measuring the evolution of cell count and extracellular metabolites (including produced carbon dioxide and consumed oxygen) and fitting the acquired data to an ordinary differential equation model. A positive value indicates metabolite production and a negative uptake. To assess the validity of the inference approach physiological rates were independently determined from unlabeled and streptavidin-labeled cell cultures (time point 0 hr), consisting of predominantly young cells. The shading reflects the inverse of the relative uncertainty of the estimation (i.e. values which are depicted with a higher transparency are more uncertain). *Figure 2—source data 2* contains the data.
DOI: https://doi.org/10.7554/eLife.41046.003

The following source data and figure supplements are available for figure 2:

**Source data 1.** Intracellular metabolite concentrations inferred for daughter and aging mother cells.
*Figure 2 continued on next page*

*Figure 2 continued*

DOI: https://doi.org/10.7554/eLife.41046.016

**Source data 2.** Growth rates and yields inferred for daughter and aging mother cells.

DOI: https://doi.org/10.7554/eLife.41046.017

**Figure supplement 1.** The cellular volume gradually increases with cellular age.

DOI: https://doi.org/10.7554/eLife.41046.004

**Figure supplement 2.** Inference of intracellular metabolite concentrations.

DOI: https://doi.org/10.7554/eLife.41046.005

**Figure supplement 3.** Comparison of inferred intracellular metabolite concentrations with independently determined concentrations of young cells.

DOI: https://doi.org/10.7554/eLife.41046.006

**Figure supplement 4.** Inference of intracellular concentrations of 18 metabolites with cell age.

DOI: https://doi.org/10.7554/eLife.41046.007

**Figure supplement 5.** The energy charge remains constant with cell age.

DOI: https://doi.org/10.7554/eLife.41046.008

**Figure supplement 6.** Inference of physiological parameters from dynamic changes in extracellular metabolites.

DOI: https://doi.org/10.7554/eLife.41046.009

**Figure supplement 7.** Inference of physiological parameters from dynamic changes in extracellular metabolites.

DOI: https://doi.org/10.7554/eLife.41046.010

**Figure supplement 8.** Inference of physiological parameters from dynamic changes in extracellular metabolites.

DOI: https://doi.org/10.7554/eLife.41046.011

**Figure supplement 9.** FBP (fructose-1,6-bisphosphate) concentration as function of sugar uptake rate.

DOI: https://doi.org/10.7554/eLife.41046.012

**Figure supplement 10.** The decreasing growth rate was confirmed using single cell analysis.

DOI: https://doi.org/10.7554/eLife.41046.013

**Figure supplement 11.** The yeast proteome progressively transforms form a fermentation- to a respiration-associated state during aging.

DOI: https://doi.org/10.7554/eLife.41046.014

**Figure supplement 12.** Replicative lifespan is increased in the presence of ethanol.

DOI: https://doi.org/10.7554/eLife.41046.015

In aging cells, we found that the specific glucose uptake rate (GUR) decreased drastically towards the end of their lifespan to almost 10% of the value of young cells (*Figure 2b*), which is in line with the simultaneously decreasing concentration of fructose-1,6-bisphosphate (*Figure 2—figure supplement 9*) and its function as a glycolytic flux-signaling metabolite (*Huberts et al., 2012*). This decrease in GUR was accompanied by a reduction of growth rate, which we qualitatively confirmed with single-cell measurements (*Figure 2—figure supplement 10*). Furthermore, while at a young age, cells showed a fermentative metabolic phenotype indicated by ethanol production and a low oxygen uptake rate (although oxygen was sufficiently available in the setup; *Janssens et al., 2015*), with increasing age cells shifted towards a respiratory phenotype as indicated by an increase in oxygen uptake and reduced ethanol excretion (*Figure 2b*). Using principle component analysis, we found a similar shift on the level of protein expression data (*Figure 2—figure supplement 11*). However, unlike a normal respiratory metabolism, where no byproducts would be excreted, up to half of the carbon influx was directed to glycerol and acetate excretion. Acetate metabolism has been linked to apoptosis (*Giannattasio et al., 2013*) and the production of glycerol indicates a stress response (*Albertyn et al., 1994*). This stress response might be crucial for survival at a high replicative age as a *gpd*1Δ (rate limiting step in the synthesis of glycerol) mutant shows a significant reduced lifespan (*Kaeberlein et al., 2002*). At the end of their lifespan (starting from time point 44 hr), cells started to co-consume ethanol, produced by surrounding daughter cells, for which we obtained independent evidence from microfluidics experiments (*Figure 2—figure supplement 12*). The identified stress responsive metabolism and decreased glucose uptake rate are consistent with signatures related to starvation and oxidative stress, as foundin our earlier proteome and transcriptome analysis (*Janssens et al., 2015*).

## Metabolic changes are accompanied by drastic intracellular flux rearrangements

To infer the normalized intracellular flux distributions (i.e. metabolic rates normalized by GUR) from the acquired physiological data, we used a recently developed computational method (*Niebel et al., 2019*). This method rests on a thermodynamic and stoichiometric model of cellular metabolism (as a function of metabolite concentration and metabolic flux) and was shown to yield predictions in good agreement with $^{13}$C based metabolic flux analysis, while not relying on labelling data (*Niebel et al., 2019*). The model consists of a mass balanced metabolic reaction network, including glycolysis, gluconeogenesis, tricarboxylic acid cycle, amino acid-, nucleotide-, sterol-synthesis and two reactions accounting for the NAD(P)H demand required for scavenging of reactive oxygen species (ROS). The reaction directionalities are constrained by the associated changes in Gibbs energy, and the Gibbs energy dissipated by the sum of all metabolic processes is balanced with the Gibbs energy exchanged with the environment through exchange processes (i.e. the production and consumption of extracellular metabolites). Using this model and regression analysis, we analysed the inferred metabolite concentrations (*Figure 2a*) and physiological rates (*Figure 2b*) (*Figure 3—figure supplement 1*). Subsequently, we assessed the solution space of the regression solution by minimizing the 'absolute sum of fluxes' (*Holzhütter, 2004*) to obtain the normalized intracellular flux distributions during aging.

The inferred intracellular metabolic rearrangements with age echo our findings from the extracellular physiology. Up until an age of 20 hr the intracellular physiology depicted a fermentative phenotype with a low normalized flux into the pentose phosphate pathway and a low normalized flux in an incomplete tricarboxylic acid cycle as the majority of carbon was leaving glycolysis through the pyruvate decarboxylase towards ethanol. After 20 hr, cells began to gradually shift towards a respiratory phenotype, where an increasing proportion of the incoming carbon flux was directed into the pentose phosphate pathway and half of the carbon flux leaving the upper glycolysis going each towards glycerol excretion and through the lower glycolysis in the tricarboxylic acid cycle, while part of the carbon loss was compensated by the uptake of ethanol and pyruvate (*Figure 3*).

This switch in metabolic operation was accompanied by an increased redox nucleotide turnover (*Figure 4*). Up until an age of 20 hr, the majority of NADH was generated in glycolysis and regenerated through the alcohol dehydrogenase. After the switch to respiration, the tricarboxylic acid cycle became the major source of NADH, which in turn was regenerated in the respiratory chain. During the first 20 hr, NADPH turnover was low but after the switch towards respiration NADPH was produced in the pentose phosphate pathway and through the aldehyde dehydrogenase. The increase in redox nucleotide turnover can be attributed to increased demands to combat emerging reactive oxygen species (ROS) (*Figure 4*). Despite these dramatic changes in cofactor turnover, cells managed to maintain a constant NAD(P)H levels, as observed in age-spanning time-lapse analysis in single cells (*Figure 4—figure supplement 1*).

## Discussion

Here, employing again the same experimental setup and procedures, we complement our earlier generated transcriptome and proteome account during the replicative aging of the budding yeast *Saccharomyces cerevisiae* (*Janssens et al., 2015*), with the metabolic phenotype, inferred from cell ensemble measurements. Next to globally decreased metabolite levels, we found that cells shift with age from a fermentative towards a respiratory phenotype accompanied by a decrease in growth and glucose uptake rate. The increase in cellular volume (and the accompanying decrease in surface area per cell volume) with age (cf. *Figure 2—figure supplement 1*) could be in part responsible for the observed decrease in the volumetric (i.e. dry weight specific) substrate influx, next to possibly altered hexose transporter expression with age (*Kamei et al., 2014*). Such decreased substrate influx will lead to decreased glycolytic fluxes, which trigger a switch towards a respiratory metabolism (*Huberts et al., 2012*). Increased respiratory activity (*Figures 2b* and *3*) could then lead to an increased generation of reactive oxygen species (*Drakulic et al., 2005*) necessitating an increase in redox cofactor turnover (*Figure 4*) for ROS scavenging. This cascade of metabolic changes, likely in part induced by the non-homeostatic volume increases and the concomitant collapse in substrate uptake rate, might not only cause detrimental effects due to for example ROS production, but the reduced metabolic rates might also be responsible for the entry into senescence, as it was recently

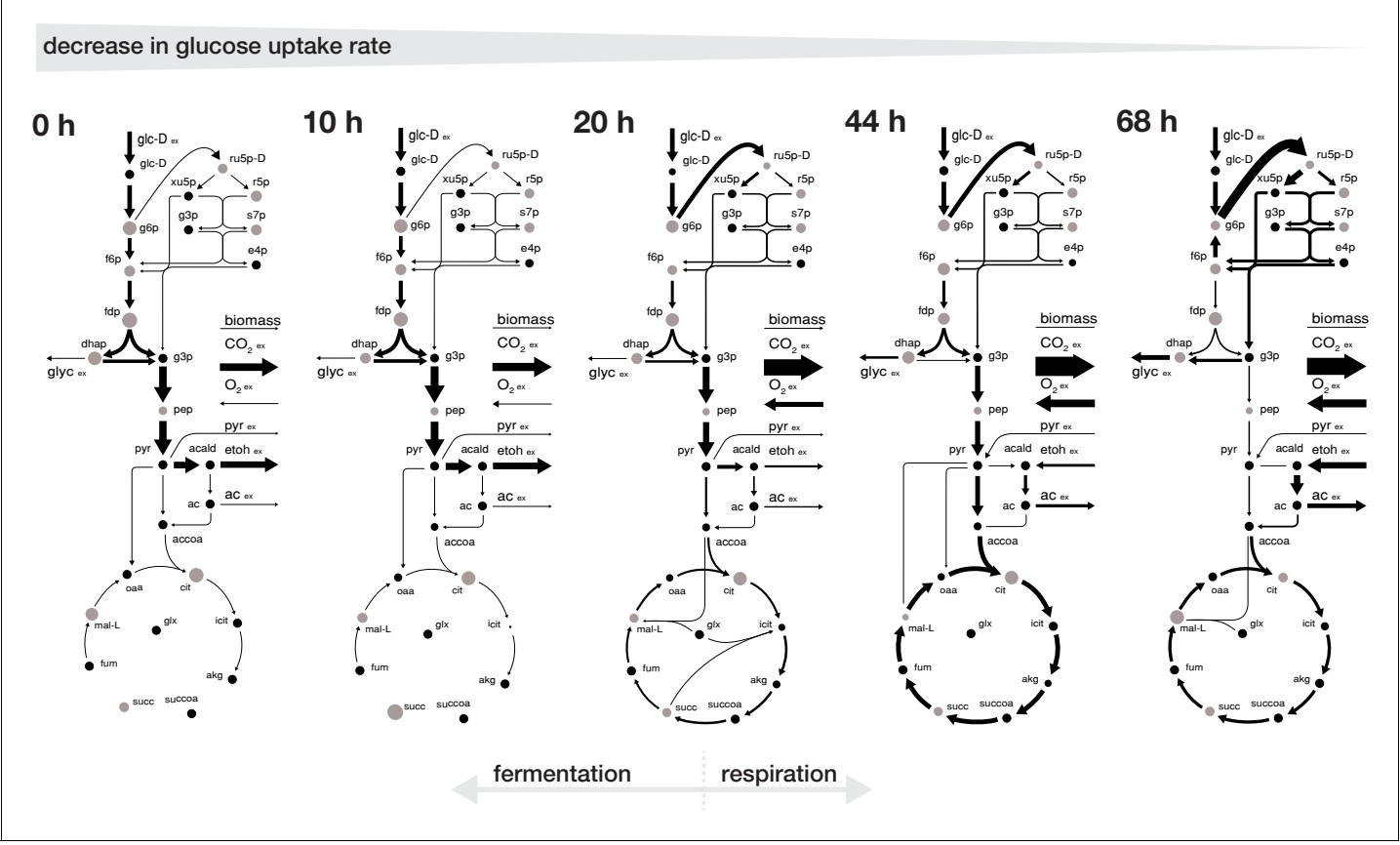

**Figure 3.** Rearrangement of normalized fluxes during replicative aging. The normalized flux distributions (i.e. metabolic rates normalized by GUR) were obtained by minimizing the 'absolute sum of fluxes' within the solution space of the regression analysis of the inferred intracellular metabolite concentrations and physiological rates. The thickness of the arrows corresponds to the absolute value of the fluxes, normalized to the glucose uptake rate. The grey dots show the intracellular metabolite concentrations inferred for cells of the respective age where the diameter corresponds to the natural logarithm of the respective concentration. Note, that this figure does not show the complete model stoichiometry of the metabolic network. The numeric values of the respective normalized fluxes can be found in *Figure 3—source data 1*.
DOI: https://doi.org/10.7554/eLife.41046.018

The following source data and figure supplement are available for figure 3:

**Source data 1.** Reaction stoichiometry of employed metabolic network model, metabolite annotation and inferred predicted intracellular metabolic fluxes normalized to the glucose uptake rates, for aging mother cells.
DOI: https://doi.org/10.7554/eLife.41046.020

**Figure supplement 1.** Results of the regression analysis using the combined thermodynamic and stoichiometric metabolic model.
DOI: https://doi.org/10.7554/eLife.41046.019

shown that sufficiently high enough metabolic rates are necessary for cells to pass cell cycle start (*Papagiannakis et al., 2017*).

## Materials and methods

### Method 1 | strain and cultivation conditions

The haploid prototrophic *Saccharomyces cerevisiae* strain, YSBN6 (MATa, FY3 ho::HphMX4) (*Canelas et al., 2010*), which is derived from S288c, was used in this study. All cultivations were performed using yeast nitrogen base (YNB) without amino acids (ForMedium, Norfolk, UK) supplemented with 2% glucose at 30°C and 300 rpm, unless indicated differently.

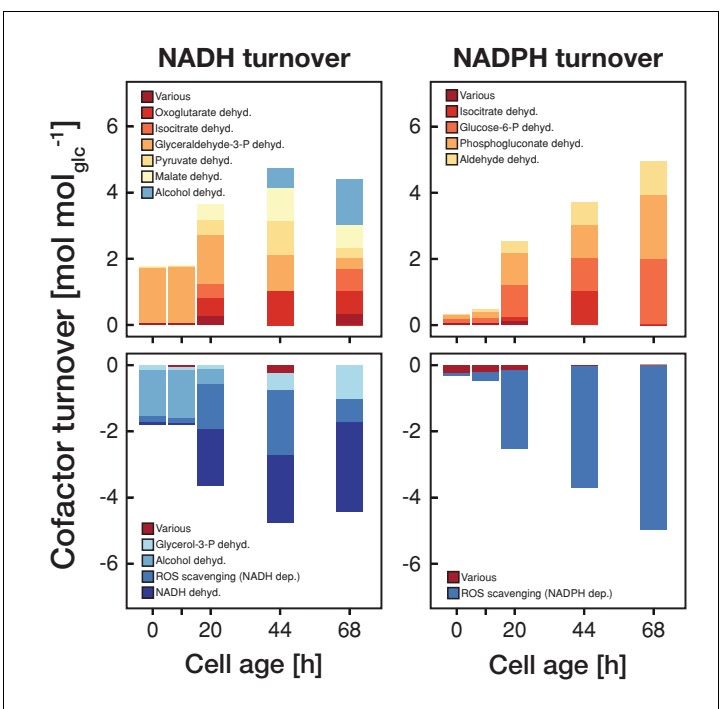

**Figure 4.** The metabolic rearrangements with age are accompanied by shifts in redox cofactor turnover. The redox cofactor production and consumption rates (normalized by the respective glucose uptake rate) were obtained by minimizing the absolute sum of fluxes within the solution space of the regression analysis of the inferred intracellular metabolite concentrations and physiological rates. Reactions with a maximal turnover of <0.5 mol $mol_{glc}^{-1}$ were combined and depicted as *various*. A positive turnover means that the cofactor is produced and a negative turnover that the cofactor is consumed. Note, that we did not enforce the emergence of ROS, however, the model could fit the experimental data the best by using cofactors for ROS scavenging.
DOI: https://doi.org/10.7554/eLife.41046.021

The following figure supplement is available for figure 4:

**Figure supplement 1.** The intracellular NAD(P)H concentration remains constant with cell age.
DOI: https://doi.org/10.7554/eLife.41046.022

## Column-based cultivation of yeast cells and sampling

To generate large quantities of aged yeast cells, necessary to perform bulk measurements, we used a method, in which cells were immobilized on iron beads and trapped inside a column (*Janssens et al., 2015*). Briefly, cells were labelled with biotin and linked to streptavidin-coated iron beads. This iron bead bound cell culture was then grown in a column, equipped with an iron grid, in which the beads (and the cells attached to them) were trapped by a magnet. A continuous medium flow through the column washed out most emerging daughter cells and kept the mother cells in a constant, nutrient-rich environment. With the used flow rate of 170 mL $h^{-1}$, the glucose concentration stayed almost constant (only dropped from 21.7 to 20.1 g $L^{-1}$) and the concentration of major byproducts (pyruvate, succinate, glycerol, acetate and ethanol) never exceeded 1 g $L^{-1}$. Furthermore, the dissolved oxygen saturation never dropped below 75%. The precise instrumental as well as experimental setup for the column-based cultivation and harvest can be found in *Janssens et al. (2015)*.

As samples harvested from the column still resembled a mixture of mother, daughter and dead cells and any subsequent sorting step, aiming at an absolutely pure mother cell fraction would have inherently led to a distortion of the metabolic phenotype, we opted for an approach also followed in our previous study (*Janssens et al., 2015*), to computationally infer the phenotype of each subpopulation. Specifically, we generated at each aging time point three samples with different proportions of mothers, daughter and dead cells (i.e. (1) from the column effluent, (2) from the column after an additional washing step, (3) from the washing solution (in the following referred to as mix 1, 2 and

3)). After harvesting and before the respective analysis (and for the physiological characterization additionally at the end of the growth experiment), the cell count specific fractional abundance of each subpopulation in each sample was determined by flow cytometry and a combined dye-staining with propidium iodide and avidin – FITC. Later the metabolite concentrations and the cellular physiologies of each individual cell population (i.e. mother, daughter and dead cells) were mathematically inferred from data originating from the mixed samples and the determined fractional abundance.

## Method 2 | inference of intracellular metabolite concentrations
### Regeneration
To allow the cells to recover from any possible stress during the sampling procedure, all samples were transferred in an Erlenmeyer flask containing 10 mL medium, adjusted to a cell density of $2 \times 10^7$ cells mL$^{-1}$ and incubated for 20 min at 30°C and 300 rpm prior analysis.

### Sample preparation
A sample of $3 \times 10^7$ cells was taken from the Erlenmeyer flask and immediately quenched in 10 mL −40°C methanol. The cells were separated from the organic solvent by centrifugation (5 min, 21'000 g, 4°C), washed with 2 mL −40°C methanol, separated again by centrifugation and stored at −80°C. For the following analysis, the cell pellet was re-suspended in 900 µL −40°C extraction buffer (methanol, acetonitrile and water, 4:4:2 v/v/v supplemented with 0.1 M formic acid) and an internal standard of $^{13}$C-labeled metabolites was added to the extraction. This standard was obtained and quantified from exponentially growing cell cultures prior to the experiment (*Wahl et al., 2014*). The extraction solution was agitated for 10 min at room temperature and thereafter centrifuged at maximum speed. The supernatant was transferred to a new vial and the cell pellet re-suspended in 900 µL −40°C extraction buffer and the extraction procedure was repeated a second time. The supernatants from both steps were combined and centrifuged for 45 min at 4°C and 21'000 g to remove any remaining non soluble parts. Thereafter, the supernatant was vacuum-dried at 45°C for approximately 1.5 hr and prior to the further analysis dissolved in 200 µL water.

### Measurement of intracellular metabolites
The extracted metabolite samples were analyzed using a UHPLC-MS/MS system. The chromatographic separation was performed on a Dionex Ultimate 3000 RS UHPLC (Dionex, Germering, Germany) equipped with a Waters Acquity UPLC HSS T3 ion pair column with precolumn (dimensions: 150 × 2.1 mm, particle size: 3 µm; Waters, Milford, MA, USA). The injection volume was 10 µL and the samples were permanently cooled at 4°C. A binary solvent gradient was employed (0 min: 100% A; 5 min: 100% A 10 min: 98% A; 11 min: 91% A; 16 min: 91% A; 18 min: 75% A, 22 min: 75% A; 22 min: 0% A; 26 min: 0% A; 26 min: 100% A; 30 min: 100% A) at a flow rate of 0.35 mL min$^{-1}$ where solvent A was composed of 5% methanol in water v/v supplemented with 10 mM tributylamine, 15 mM acetic acid and 1 mM 3,5-heptanedione and isopropanole as solvent B. The detection was done using multiple reaction monitoring (MRM) on a MDS Sciex API365 tandem mass spectrometer, upgraded to EP10+ (Ionics, Bolton, Ontario, Canada) and equipped with a Turbo-Ionspray source (MDS Sciex, Nieuwerkerk aan den Ijssel, Netherlands) with the following source parameter: NEB (nebulizing gas, N2): 12 a.u., CUR (curtain gas, N2): 12 a.u., CAD (collision activated dissociation gas): 4 a.u., IS (ion spray voltage): −4,500 V, TEM (temperature): 500°C.

### Mathematical inference of intracellular metabolite concentrations of mother and daughter cells using non-negative least squares regression
The concentrations of intracellular metabolites were determined from samples harvested after 10, 20, 44, and 68 hr. The samples were measured in six replicates and the average of this replicates was used for the mathematical inference. To validate the interference approach we independently determined the intracellular metabolite concentrations of biotin labeled cells before loading them onto the column.

The general idea of the in the following described mathematical inference rests on the concept that a system of linear equations can be solved if the number of independent equations is greater or equal than the number of unknowns. This was implemented by generating at each time point three samples (i.e. mix 1, 2 and 3, cf. Methods 1). The measured concentration in each of these three

samples is constituted as the sum of the two unknown concentrations in mother and daughter cells, weighted by their respective known fractional abundance.

Specifically, the in each sample (with $n^{cell}$ cells) measured amount of metabolite, $n^{meas}$, contains metabolites originating from mother ($mo$) and daughter ($da$) cells. As dead cells were considered to be lysed and their metabolite content accordingly leaked into the medium, we assumed that their contribution to the total metabolite pool can be neglected. With taking the respective volumes of mother and daughter cells (Method 5 and *Figure 2—figure supplement 1*), and the fractional abundance of each population into account, the amount of substance of each metabolite in each cell is given by,

$$\frac{n^{meas}_{i,j,k}}{n^{cell}_{j,k}} = \alpha_{j,k} V^{mo}_k c^{mo}_{i,k} + \beta_{j,k} V^{da} c^{da}_i, \tag{1}$$

where $n^{meas}_{i,j,k}$ is the measured amount of substance (unit mol) of the metabolite $i$ in the sample $j$ (i.e. mix 1, 2 or 3) at the aging time point $k$ (i.e. 10, 20, 44 or 68 hr), $n^{cell}_{j,k}$ the total amount of cells in the respective sample, $\alpha_{j,k}$ and $\beta_{j,k}$ the cell count specific fractional abundance of mother and daughter cells, $V^{mo}_k$ and $V^{da}$ the cell volume (unit L cell$^{-1}$) of mother and daughter cells and $c^{mo}_{i,k}$ and $c^{da}_i$ the unknown metabolite concentration (unit M) in mother and daughter cells. Note that $c^{da}_i$ and $V^{da}$ are not indexed over the aging time points $k$, as we assumed that the daughter cell phenotype does not change over time (i.e. daughter cells produced by young mothers are identical with daughter cells produced by old mothers). To infer the intracellular metabolite concentrations $c^{mo}$ and $c^{da}$ from the measurements, $n^{meas}$, we formulated a non-negative least square regression problem of the form,

$$\min_x \|Ac - n\|^2_2, c \geq 0, \tag{2}$$

where the matrix $A$ contains all fractional volumes $\alpha_{j,k} V^{mo}_k$ and $\beta_{j,k} V^{da}$ in every sample $j$ at every aging time point $k$, the vector $c$ the unknown concentrations $c^{mo}_{i,k}$ and $c^{da}_i$ of the metabolite $i$ in mother and daughter cells at every aging time point $k$ and the vector $n$ all metabolite measurements, $n^{meas}_{i,j,k}$, normalized by the total amount of cells in the sample, $n^{cell}_{j,k}$, in every sample $j$ at every aging time point $k$.

The regression problem in *Equation 2* was implemented in MATLAB (Release R2013, MathWorks, Inc, Massachusetts, USA) and the unknown metabolite concentrations, $c$, in mother and daughter cells were identified using the function 'lsqnonneg'. The uncertainty of the estimation was then determined by leave-one-out cross-validation, where we one-by-one removed data points from the set and repeated the estimation procedure (*Figure 2—figure supplement 4*).

## Method 3 | inference of growth, metabolite uptake and production rates

The physiological parameters (i.e. growth, metabolite uptake and production rates) were determined from two independent experimental campaigns. In campaign I, samples were harvested after 20, 44 and 68 hr and in campaign II after 10, 20, 44, and 68 hr where the samples from campaign II were split and analyzed in duplicates. The three data sets of both campaigns were combined for the inference. Additionally, we determined the physiologies of biotin labeled cells (referred to as '0 hr') and unlabeled cells (referred to as 'unlabeled').

### Batch cultivation conditions in minimal medium

The three samples obtained from the cultivation column (i.e. mix 1, 2 and 3) as well as the two reference samples (i.e. 0 hr and unlabeled) were transferred each in a 250 mL Erlenmeyer flask (or RAMOS flasks) containing 25 mL medium, adjusted to a cell density of $2 \times 10^7$ cell mL$^{-1}$, and incubated at 300 rpm and 30°C.

### Determination of cell dry weight from cell count

The cell count was measured every 20 min between 1 and 3 hr after inoculation using a BD Accuri C6 flow cytometer (Becton, Dickinson and Company, Franklin Lakes, NJ). The samples were diluted with PBS at pH seven to <10$^6$ cell mL$^{-1}$ and 20 µL sample were counted at 'medium' flow. The

FSC-H thresholds was set to 80'000 in order to cut off most of the electronic noise. To correct the measured dry weight for the mass of iron beads in the sample, the iron beads were gated separately and counted as well. The data were analyzed using the Accuri CFlow Plus software.

As the cell volume and thus the cell specific dry weight (i.e. the weight of one cell) of mother cells changes with age, towards converting the measured cell counts to dry weight (biomass), we first determined the cell specific dry weight of mother/dead, $m^{mo/de}$, and daughter cells, $m^{da}$. After 3 hr, at the end of each batch cultivation, 20 mL of culture were filtered through a pre-weighed nitrocellulose filter with a pore size of 0.2 µm. The filter was washed once with distilled water, dried at 80°C for two days and afterwards weighed again. The total weight of iron beads attached to mother cells (here we assumed that one mother cell is attached to one iron bead; *Janssens et al., 2015*) and free beads, which was determined from the counted number of iron beads in the sample and the weight of one individual bead, was subtracted from the total dry weight of each sample. The bead weight had been determined to be $8.49 \times 10^{-13}$ g per bead by filtration and weighting of a known amount of beads. Next, the cell specific dry weight of mother/dead and daughter cells was inferred from the measured population-average dry weight in the samples, $m^{meas}$, by following an in principle similar approach as done for the intracellular metabolite concentrations. Specifically, we assumed that dead cells (i.e. died mother cells) and mother cells have the same dry mass and that the dry mass of newly formed daughter cells does not change over the aging time points. Taking the fractional abundances of each cell population into account, the measured cell specific dry mass in each sample is given as,

$$\frac{m^{meas}_{j,k}}{n^{cell}_{j,k}} = \left(\alpha_{j,k} + \gamma_{j,k}\right)m^{mo/de}_{k} + \beta_{j,k}m^{da}, \tag{3}$$

where $m^{meas}_{j,k}$ is the measured population-average dry mass (unit g) after 3 hr cultivation in the sample $j$ at the aging time point $k$, $n^{cell}_{j,k}$ the total amount of cells in the respective sample, $\alpha_{j,k}$ the cell count specific fraction of mother cells, $\gamma_{j,k}$ the cell count specific fraction of dead cells, $m^{mo/de}_{k}$ the unknown cell specific dry mass (unit g) of mother or dead cells, $\beta_{j,k}$ the cell count specific fraction of daughter cells and $m^{da}$ the unknown cell specific dry mass (unit g) of daughter cells. Next, we formulated a least square regression problem of the form,

$$\min_{x}\|Am - n\|^2_2, \tag{4}$$

where the matrix $A$ contains all fractional abundances $\alpha_{j,k} + \gamma_{j,k}$ and $\beta_{j,k}$ in every sample $j$ at every aging time point $k$, the vector $m$ the unknown cell specific dry weights $m^{mo/de}_{k}$ and $m^{da}$ at every aging time point $k$ and the vector $n$ all measured cell dry weights, $m^{meas}_{j,k}$, normalized by the total amount of cells in the sample, $n^{cell}_{j,k}$, in every sample $j$ at every aging time point $k$. The regression problem in *Equation 4* was implemented in R (Release 3.2.0) and the unknown cell specific dry weights, $m$, of mother/dead and daughter cells were identified using the function 'lm'.

The inferred cell specific dry weights of mother/dead and daughter cells were then used to convert the measured cell counts to dry weight. At the beginning of each cultivation ($t = 0$) the total dry weight, $Xt_{t=0}$, is constituted of mother/dead and daughter cells, taking their fractional abundance into account, while in the following all new emerging cells are daughter cells. The total dry weight at every time $t$, $X_t$, is then given as,

$$X_{t,j,k} = \underbrace{\left(\alpha_{t=0,j,k} + \gamma_{t=0,j,k}\right)n_{t=0,j,k}m^{mo/de}_{k} + \beta_{t=0,j,k}n_{t=0,j,k}m^{da}}_{X_{t=0,j,k}} + \left(n_{t,j,k} - n_{t=0,j,k}\right)m^{da}, \tag{5}$$

where $X_{t,j,k}$ is the dry weight of the mixed population sample $j$ of the aging time point $k$ at time $t$, $\alpha_{t=0,j,k} + \gamma_{t=0,j,k}$ and $\beta_{t=0,j,k}$ the cell count specific fractional abundances of mother/dead and daughter cells at the beginning of the cultivation, $n_{t=0,j,k}$ the cell count at the beginning of the cultivation and $n_{t,j,k}$ the cell count at the time $t$. Note that $k$ refers to the cell age (i.e. aging time point) and $t$ refers to the cultivation time at each aging time point (between 0 and 3 hr).

Additionally, the inferred cell specific dry weights of mother/dead and daughter cells were used to convert the cell count specific fractional abundances, $\alpha_{j,k}$, $\beta_{j,k}$, and $\gamma_{j,k}$, in the dry mass specific fractional abundances of mother, daughter and dead cells, $\alpha^{dw}_{j,k}$, $\beta^{dw}_{j,k}$, and $\gamma^{dw}_{j,k}$, in every sample $j$ at every aging time point $k$:

$$\alpha_{j,k}^{dw} = \frac{\alpha_{j,k} m_k^{mo/de}}{(\alpha_{j,k} + \gamma_{j,k}) m_k^{mo/de} + \beta_{j,k} m^{da}}, \tag{6}$$

$$\beta_{j,k}^{dw} = \frac{\beta_{j,k} m_k^{mo/de}}{(\alpha_{j,k} + \gamma_{j,k}) m_k^{mo/de} + \beta_{j,k} m^{da}}, \tag{7}$$

$$\gamma_{j,k}^{dw} = \frac{\gamma_{j,k} m_k^{mo/de}}{(\alpha_{j,k} + \gamma_{j,k}) m_k^{mo/de} + \beta_{j,k} m^{da}}, \tag{8}$$

## Determination of glucose and extracellular metabolite concentration

0.3 mL samples were taken every 20 min from 1 to 3 hr after inoculation. To separate the cells from the medium, the samples were centrifuged at maximum speed for 3 min, the supernatant transferred onto a filter column (SpinX, pore size 0.22 μm), again centrifuged at maximum spend and the flow through was further analyzed. The glucose, pyruvate, glycerol, acetate and ethanol concentration was detected using an Agilent 1290 LC HPLC system equipped with a Hi-Plex H column and 5 mM $H_2SO_4$ as eluent at a constant flow rate of 0.6 mL min$^{-1}$. The injection volume was 10 μL and the column temperature was kept constant at 60°C. Glucose, glycerol, ethanol and acetate were detected by refractive index and pyruvate by UV (constant wave length of 210 nm) and the respective concentrations were determined using an external standard with known concentrations. The data were analyzed using the Agilent Open Lab CDS software.

## Determination of total consumed oxygen and produced carbon dioxide

The oxygen transfer rate (OTR) and carbon dioxide transfer rate (CTR) were determined from exhaust gas analysis using a respiration activity monitoring system (RAMOS) (*Hansen et al., 2012*). The RAMOS measurement flask, containing 25 mL medium, was inoculated with $2 \times 10^7$ cell mL$^{-1}$ and the cultivation conditions were identical to the batch cultures used to determine the other physiological parameters. One RAMOS measurement cycle encompassed a 10 min measuring phase and a 20 min rinsing phase. The total consumption of oxygen and the production of carbon dioxide in a time interval were calculated from the mean of two consecutive OTR and CTR measurement cycles multiplied by the time.

## Inference of growth, metabolite uptake and production rates of mother and daughter cells

To infer the physiological parameter of mother (*mo*), daughter (*da*) and dead (*de*) cells from the mixed population measurements, we formulated an ordinary differential equation model describing the dynamic change of biomass and extracellular metabolites during the 3 hr cultivation in each sample. To this end, we assumed that the physiology of daughter cells stays constant over all aging time points and that within the 3 hr cultivation the physiology of the mother cells stays constant. Finally, due to the short experiment time the evaporation of water and metabolites was neglected.

The total biomass in the sample is constituted of mother, dead and daughter cells and thus the differential mass balance can be formulated as,

$$0 = \frac{d}{dt}\alpha_{j,k}^{dw} + \frac{d}{dt}\beta_{j,k}^{dw} + \frac{d}{dt}\gamma_{j,k}^{dw}. \tag{9}$$

Due to the short experiment time (3 hr) compared to their life span (>50 hr), we assumed that the amount of initial mother and dead cells stays constant (i.e. no new mother cells emerge and no mother cells die during the experiment). Thus,

$$\frac{d}{dt}X_{j,k}^{mo} = \frac{d}{dt}\left(\alpha_{j,k}^{dw} X_{j,k}\right) = 0, \tag{10}$$

and

$$\frac{d}{dt}X_{j,k}^{de} = \frac{d}{dt}\left(\gamma_{j,k}^{dw}X_{j,k}\right) = 0, \tag{11}$$

where $X_{j,k}$ is the total biomass and $X_{j,k}^{mo}$ and $X_{j,k}^{de}$ the biomass of mother and dead cells in sample $j$ at the aging time point $k$.

From *Equation 9, 10 and 11*, and follows that the change in total biomass is only due to the change in daughter cell biomass, $X_{j,k}^{da}$, which in turn can be either due to the emergence of new daughter cells originating from mother cells (i.e. budding of mother cells) or originating from daughter cells (i.e. budding of daughter cells). Thus, the change of the total biomass is given as,

$$\frac{d}{dt}X_{j,k} = \frac{d}{dt}X_{j,k}^{da} = \frac{d}{dt}\left(\beta_{j,k}^{dw}X_{j,k}\right) = \mu_k^{mo}\alpha_{j,k}^{dw}X_{j,k} + \mu^{da}\beta_{j,k}^{dw}X_{j,k}, \tag{12}$$

where $\mu_k^{mo}$ is the growth rate (unit $h^{-1}$) of mother cells and $\mu^{da}$ is the growth rate (unit $h^{-1}$) of daughter cells.

Reformulating the partial derivatives in *Equations 10 and 11* and adding *Equation 12* yields the change in dry mass specific fractional abundance of mother and dead cells as,

$$\frac{d}{dt}\alpha_{j,k}^{dw} = \frac{\alpha_{j,k}^{dw}}{X_{j,k}}\frac{d}{dt}X_{j,k} = -\alpha_{j,k}^{dw}\left(\alpha_{j,k}^{dw}\mu_k^{mo} + \beta_{j,k}^{dw}\mu^{da}\right), \tag{13}$$

and

$$\frac{d}{dt}\gamma_{j,k}^{dw} = \frac{\gamma_{j,k}^{dw}}{X_{j,k}}\frac{d}{dt}X_{j,k} = -\gamma_{j,k}^{dw}\left(\alpha_{j,k}^{dw}\mu_k^{mo} + \beta_{j,k}^{dw}\mu^{da}\right), \tag{14}$$

and plugging *Equations 13 and 14* and in the differential biomass balance (*Equation 9*) yields the change in fractional abundance of daughter cells due to budding of mother and daughter cells as,

$$\frac{d}{dt}\beta_{j,k}^{dw} = \left(\alpha_{j,k}^{dw} + \gamma_{j,k}^{dw}\right)\left(\alpha_{j,k}^{dw}\mu_k^{mo} + \beta_{j,k}^{dw}\mu^{da}\right). \tag{15}$$

Next, the change in glucose concentration in the medium can be due to the uptake by mother and daughter cells as in,

$$\frac{d}{dt}c_{glc,j,k} = -X_{j,k}\left(\alpha_{j,k}^{dw}\underbrace{\frac{\mu_k^{mo}}{Y_{XS,k}^{mo}}}_{q_{S,k}^{mo}} + \beta_{j,k}^{dw}\underbrace{\frac{\mu^{da}}{Y_{XS}^{da}}}_{q_S^{da}}\right), \tag{16}$$

where $c_{glc,j,k}$ is the measured glucose concentration (unit $g\ L^{-1}$) in sample $j$ at the aging time point $k$, $q_{S,k}^{mo}$ and $q_S^{da}$ the specific uptake rates of mother and daughter cells and $Y_{XS,k}^{mo}$ and $Y_{XS}^{da}$ the biomass yields (unit $g\ g_{GLU}^{-1}$) of mother and daughter cells.

In a similar way, the mass balance for oxygen, carbon dioxide and other fermentation products can be formulated:

$$\frac{d}{dt}c_{O_2,j,k} = -X_{j,k}\left(\alpha_{j,k}^{dw}\underbrace{Y_{O_2S,k}^{mo}\frac{\mu_k^{mo}}{Y_{XS,k}^{mo}}}_{q_{O_2,k}^{mo}} + \beta_{j,k}^{dw}\underbrace{Y_{O_2S}^{da}\frac{\mu^{da}}{Y_{XS}^{da}}}_{q_{O_2}^{da}}\right), \tag{17}$$

$$\frac{d}{dt}c_{P,j,k} = X_{j,k}\left(\alpha_{j,k}^{dw}\underbrace{Y_{PS,k}^{mo}\frac{\mu_k^{mo}}{Y_{XS,k}^{mo}}}_{q_{P,k}^{mo}} + \beta_{j,k}^{dw}\underbrace{Y_{PS}^{da}\frac{\mu^{da}}{Y_{XS}^{da}}}_{q_P^{da}}\right), \tag{18}$$

where $q_{O2,k}^{mo}$, $q_{O2}^{da}$, $q_{P,k}^{mo}$ and $q_P^{da}$ are the biomass specific oxygen uptake and product (including

carbon dioxide) excretion rates (unit g $g_{DW}^{-1}$ $h^{-1}$) of mother and daughter cells at the aging time point $k$ and $Y_{O2S}{}^{mo}{}_k$, $Y_{O2S}{}^{da}$, $Y_{PS}{}^{mo}{}_k$ and $Y_{PS}{}^{da}$ the respective oxygen and product yields (unit g $g_{GLU}{}^{-1}$) of mother and daughter cells.

To increase robustness in the estimation, we stated that the mother and daughter cell physiology needs to fulfill the carbon balance within a certain range.

$$0.5 \leq \frac{\sum q_P^C}{q_S^C} \leq 1.5, \tag{19}$$

where $q_S^C$ and $q_P^C$ are the specific carbon uptake and excretion rates (unit C-mol $g_{DW}^{-1}$ $h^{-1}$) of mother and daughter cells.

All three datasets were combined into one parameter estimation problem subject to the *Equations 12–19*. All parameters (including initial conditions) as well as the associated uncertainties were estimated using Maximum Likelihood estimation implemented in the software gPROMS Model-Builder (Release 4.0, PSE software systems) with the MINLP solver SRQPD where a constant variance (error model) was assumed for all measurements.

## Method 4 | inference of intracellular metabolic fluxes

### Computational model of cellular metabolism

To determine the intracellular fluxes at different cell ages from the inferred metabolite concentrations and physiologies, we made use of a recently published computational inference method (*Niebel et al., 2019*). This method rests on a combined thermodynamic and stoichiometric network model of cellular operation, $M(v,\ln c) \leq 0$ (*Equation 20*), consisting of a mass balanced metabolic reaction network, in which the reaction directionalities are constraint by the associated changes in Gibbs energy – as a function of the metabolite concentrations $c$ – through the 2nd law of thermodynamics. Additionally, the Gibbs energy, which is dissipated through metabolic operation (i.e. the sum of all metabolic processes, *MET*) is balanced with the Gibbs energy exchanged with the environment through exchange processes (i.e. the production and consumption of metabolites, *EXG*),

$$\{M(v, \ln c) \leq 0\} = \left\{ \begin{array}{c} \sum_{j \in MET} S_{ij} v_j = v_{i \in EXG} \ \forall i \\ \Delta_r G'(\ln c_j) v_j \leq 0 \ \forall j \in MET \\ \sum_{j \in MET} \Delta_r G'(\ln c_j) v_j = \sum_{i \in EXG} \Delta_f G'(\ln c_i) v_i \end{array} \right\}, \tag{20}$$

where $S_{ij}$ is the stoichiometric coefficient of the $i^{th}$ reactant (i.e. metabolite) in reaction $j$, $v_j$ the rate of the reaction $j$ (i.e. the flux through this reaction), $\Delta_r G'(\ln c_j)$ the Gibbs free energy of reaction of the metabolic process $j$ and $\Delta_f G'(\ln c_i)$ the Gibbs free energy of formation of the reactant $i$.

The published, and here used, model for *Saccharomyces cerevisiae* encompasses the metabolic processes of glycolysis, gluconeogenesis, tricarboxylic acid cycle, amino acid-, nucleotide-, sterol-synthesis and considers the processes' location in the cytosol, mitochondria and extracellular space. To account for cofactor turnover due to the combatting of reactive oxygen species, which is known to occur at high replicative ages (*Ayer et al., 2014*), the model was extended by reactions describing the oxidation of NADH and NADPH through glutathione in the cytoplasm as well as the glutathione exchange (i.e. a sink and a source). This exchange does not represent any direct metabolic process but needed to be included since the glutathione metabolism is not part of this model.

```
nadh[c] + gthox[c] => nad[c] + (2) gthrd[c]
nadh[c] + gthox[c] => nad[c] + (2) gthrd[c]
          gthox[c] <=>
          gthrd[c] <=>
```

A more detailed description of this model and its implementation can be found in *Niebel et al. (2019)*.

### Regression analysis

Using this model and the inferred age-dependent metabolite concentrations and physiologies, we formulated a regression problem minimizing the weighted residual sum of squares, *RSS(y)*

(*Equation 21*). As data we used (i) the inferred yields, $\tilde{Y}_i^{(k)}$ ($i \in PY$... physiological yield), (ii) the inferred metabolite concentrations $\tilde{c}_i^{(k)}$ ($i \in MC1 \bigcup i \in MC2$... metabolite concentration set 1 or 2 (see below)), both of daughter and aged mother cells at a replicate age of 0, 10, 20, 44 and 68 hr and (iii) standard Gibbs energies of reaction, $\Delta_r \tilde{G}_j'^o$. The later were determined (including uncertainty) using the component contribution method (*Noor et al., 2013*) and as this was not possible for all standard Gibbs energies, to prevent overfitting, the regression was regularized by the Lasso method (*Hastie et al., 2011*).

To ensure the same thermodynamic reference state (i.e. the same standard Gibbs energies of reactions) in all experimental conditions, we bundled all datasets in on regression problem and indexed the model (*Equation 20*) over the experimental conditions k.

$$
\begin{aligned}
\overline{RSS}(y) \quad =& \frac{1}{\#n_Y} \sum_{k,i \in PY} \left( \frac{\frac{v_i^{(k)}}{v_{glc-D\_EX}^{(k)}} - \tilde{Y}_i^{(k)}}{\tilde{Y}_i^{(k),SE}} \right)^2 \\
&+ \frac{1}{\#n_c} \left[ \sum_{k,i \in MC1} \left( \frac{e^{\ln c_{i[c]}^{(k)}} - \tilde{c}_i^{(k)}}{\tilde{c}_i^{(k),SE}} \right)^2 + \sum_{k,i \in MC2} \left( \frac{0.9e^{\ln c_{i[c]}^{(k)}} + 0.1e^{\ln c_{i[m]}^{(k)}} - \tilde{c}_i^{(k)}}{\tilde{c}_i^{(k),SE}} \right) \right], \\
&+ \frac{1}{\#n_{CCM}} \sum_{j \in CC} \left( \frac{\Delta_r G_j'^o - \Delta_r \tilde{G}_j'^o}{\Delta_r \tilde{G}_j'^{o,SE}} \right)^2 + \frac{0.05}{\#n_{unk}} \left| \Delta_r G_j'^o \right|
\end{aligned}
\tag{21}
$$

where $\#n_Y$ and $\#n_c$ are the number of inferred yields and metabolite concentrations, $\#n_{CCM}$ the number of standard Gibbs energies of reaction, which could be estimated by the component contribution method and $\#n_{unk}$ the number of reactions where no standard Gibbs energy of reaction could be calculated. The residuals were weighted by the respective prediction uncertainty, indicated by the superscript SE. Metabolites can occur in the cytoplasm and/or in the mitochondrial space (MC1... metabolites occurring in one compartment and MC2... metabolites occurring in two compartments). Thus, we stated that the sum of the metabolite concentrations in the respective compartments, weighted by the fractional compartmental volume (0.9 for the cytoplasm and 0.1 for the mitochondrial space), had to be equal to the inferred (cell-averaging) concentration. Last, to facilitate the convergence of the optimization and for an easy conversion of reaction rates to yields, the glucose uptake rate, $v_{glc-D\_EX}$, was constraint to a value of 1 mmol gDW$^{-1}$ h$^{-1}$.

The regression analysis was implemented in the mathematical programming system GAMS (GAMS Development Corporation; General Algebraic Modeling System (GAMS) Release 24.2.2. Washington, DC, USA).

## Evaluation of the solution space

To obtain a picture of the intracellular flux distribution, we formulated the solution space, $\Omega^{reg}$ (*Equation 22*), of the optimal regression solution, indicated by an *,

$$
\begin{aligned}
\Omega^{reg} \quad =& \Big\{ \big( v^{(k)}, \ln c^{(k)}, \Delta_r G'^o \big) \big| M^{(k)} \big( v^{(k)}, \ln c^{(k)}, \Delta_r G'^o \big) \\
&\wedge \left( \frac{v_i^{(k)}}{v_{glc-D\_EX}^{(k)}} = Y_i^{(k)*} \forall i \in PY \right) \\
&\wedge \Big( \ln c_i^{(k)} = \ln c_i^{(k)} \forall i \in MC1 \Big) \wedge \Big( 0.9e^{\ln c_{i[c]}^{(k)}} + 0.1e^{\ln c_{i[m]}^{(k)}} = 0.9e^{\ln c_{i[c]}^{(k)*}} + 0.1e^{\ln c_{i[m]}^{(k)*}} \forall i \in MC2 \Big) \\
&\wedge \big( \Delta_r G'^o = \Delta_r G'^{o*} \big) \Big\}
\end{aligned}
\tag{22}
$$

Within this solution space we then minimized the 'absolute sum of fluxes',

$$
min \left\{ \sum_j |v_j| : (v, \ln c) \in \Omega^{reg} \right\}.
\tag{23}
$$

The optimization problem in *Equation 23* was implemented in the mathematical programming system GAMS (GAMS Development Corporation; General Algebraic Modeling System (GAMS) Release 24.2.2. Washington, DC, USA).

## Method 5 | determination of NAD(P)H concentration, budding rate, cell size and replicative lifespan using single cell analysis

### Microscopy

For microscopy experiments, cells from exponentially growing batch cultures were used to load a microfluidic device (*Huberts et al., 2013*; *Lee et al., 2012*). Individual cells were monitored using an inverted fluorescence microscope (Eclipse Ti-E; Nikon) housed in an custom-made microscope incubator (Life Imaging Services GmbH) that retained at a constant temperature of 30°C. During the experiment, cells were continuously fed with fresh medium. An LED-based excitation system (pE2; CoolLED) was used for illumination, and images were recorded using an Andor 897 Ultra EX2 EM-CCD camera. NAD(P)H autofluorescence (excitation at 365 nm using a 357/44 nm filter and a 409 nm beam-splitter, 200 ms exposure time, 15 % light intensity, 435/40 nm emission, EM gain 1) was recorded every 60 min to minimize phototoxic effects, and brightfield images every 10 min to reliably track individual cells and determine their division times. A CSI S Fluor 40x Oil (NA = 1.3; Nikon) objective was used for NAD(P)H. Automated hardware (PFS, Nikon) was used for correction of axial focus fluctuations during imaging.

### Image and data analysis

Cell segmentation for estimation of cell volume and fluorescence intensity took place in a semi-automated manner using the ImageJ plugin BudJ (*Ferrezuelo et al., 2012*). For cell volume estimation, brightfield images captured with the 60x objective were used. Fluorescent intensity measurements were corrected for background fluorescence using the Rolling Ball Radius algorithm of ImageJ. For budding rate estimations on the basis of single-cells, the doubling time, td, (time from bud emergence to bud emergence) was measured for each cell in 60x brightfield images, and the budding rate for each doubling event ($\ln(2)\, t_d^{-1}$) was calculated.

### Replicative lifespan

Cells from an exponentially growing culture (minimal medium; *Verduyn et al., 1992*) supplemented with 1 % (w/v) glucose were loaded in two identical microfluidic devices located on one cover glass. Minimal media supplemented with 0.5 % (w/v) glucose with and without 0.1 % (v/v) ethanol were constantly supplied into the two microfluidic devices, respectively. The cells in the microfluidic devices were monitored simultaneously by taking bright-field images every 10 minutes for more than 5 days (halogen lamp with a UV-blocking filter, 60x objective). The time points of budding, death and washout loss were recorded for individual cells using a custom macro in ImageJ. The number of budding events and fate (death or washed) of the individual cells in both microfluidic devices were used to assess the replicative age-associated survival via the Kaplan-Meier estimator. The analysis was implemented using the Lifelines (0.9.4) module in Python (2.7.13). The mean survival and its standard error were calculated using the Survival (2.43-3) package in R (3.4.1) integrating the survival curves until 44 buds (the maximal number of buds per cell in two conditions).

## Acknowledgements

We thank Silke Vedelaar for support during the metabolite extraction, Pieter Schmal and Alfredo Ramos from Process Systems Enterprise (PSE) for their support on the implementation of the ODE model.

## Additional information

### Funding

| Funder | Grant reference number | Author |
|---|---|---|
| Nederlandse Organisatie voor Wetenschappelijk Onderzoek | | Matthias Heinemann |
| European Commission | 642738 | Vakil Takhaveev<br>Matthias Heinemann |

The funders had no role in study design, data collection and interpretation, or the decision to submit the work for publication.

### Author contributions
Simeon Leupold, Software, Formal analysis, Investigation, Visualization, Methodology, Writing—original draft, Writing—review and editing; Georg Hubmann, Conceptualization, Software, Formal analysis, Investigation, Visualization, Writing—original draft, Writing—review and editing; Athanasios Litsios, Vakil Takhaveev, Formal analysis, Investigation, Visualization; Anne C Meinema, Conceptualization, Investigation; Alexandros Papagiannakis, Georges Janssens, David Siegel, Investigation; Bastian Niebel, Methodology; Matthias Heinemann, Conceptualization, Supervision, Funding acquisition, Writing—original draft, Project administration, Writing—review and editing

### Author ORCIDs
Simeon Leupold (iD) https://orcid.org/0000-0002-7186-7061
Athanasios Litsios (iD) http://orcid.org/0000-0003-3588-4988
Anne C Meinema (iD) https://orcid.org/0000-0002-0002-3486
Vakil Takhaveev (iD) http://orcid.org/0000-0002-3474-5241
Alexandros Papagiannakis (iD) http://orcid.org/0000-0002-6363-804X
Matthias Heinemann (iD) http://orcid.org/0000-0002-5512-9077

### Decision letter and Author response
Decision letter https://doi.org/10.7554/eLife.41046.026
Author response https://doi.org/10.7554/eLife.41046.027

## Additional files

### Supplementary files
• Transparent reporting form
DOI: https://doi.org/10.7554/eLife.41046.023

### Data availability
All data generated or analysed during this study are included in the manuscript and supporting files.

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
