## [Decision Letter]

Thank you for submitting your article "*Saccharomyces cerevisiae* goes through distinct metabolic phases during its replicative lifespan" for consideration by *eLife*. Your article has been reviewed by two peer reviewers, and the evaluation has been overseen by a Reviewing Editor and Naama Barkai as the Senior Editor. The reviewers have opted to remain anonymous.

The reviewers have discussed the reviews with one another and the Reviewing Editor has drafted this decision to help you prepare a revised submission.

Summary:

In this Research Advance, Leupold and colleagues take advantage of a previous method that their lab has developed to isolate aged yeast cells with the goal to analyze changes that occur in the metabolome during the ageing process. The major finding is that cells undergo a metabolic shift from fermentation to respiration during aging. The reviewers agreed that this is an interesting study that in principle is suitable for *eLife*. However, additional experiments and edits to the text will be needed before the paper can be considered for publication.

Essential revisions:

1) The paper is largely descriptive, without getting too much into causality. Some simple tests should be performed to either refute or support some of the authors' hypotheses. For example, to address the role of glycerol, what is the replicative lifespan of mutants that are impaired in glycerol synthesis (e.g., gpd1 mutants)?

2) The authors use a series of complicated mathematical models to indirectly infer the metabolite level, physiological parameters and intracellular metabolic fluxes at different ages from the raw data. Given that quite a number of assumptions and approximations have been used in their models, it is hard to fully trust the inferred results without other supporting evidence. Additional controls and further validation will be needed: (a) Unlabeled cells cultured in column with non-fermentable carbon source would serve as a good control for the respiratory state and can be used to reflect the significance of the changes during aging. (b) Use another population-level aging approach, such as Mother-Enrichment Program, to reproduce some of the metabolite data. It will not be necessary to reproduce all the data, but a couple of important metabolites at a few time points would be important.

3) In Figure 2 and Figure 2—figure supplement 2C, all the metabolite levels drop sharply within 0 – 10 hour (the first time point collected from the column). This is surprisingly early in the aging process. In addition, even some of the physiological parameters, such as the growth rate, drop drastically at 10-hour time point (Figure 2B). This is strikingly different from the time trace they obtained using microfluidics, where there is no change in the first 20 hours (Figure 2—figure supplement 5). This discrepancy raises the suspicion that these metabolic changes may be simply a part of the stress response induced by the column-based culturing condition. Additional controls would be needed to confirm that the column-based culturing condition is not stressful, e.g. using stress responsive genes that will not be induced during the first 10 hours of aging in microfluidics. Ideally, one would hope to see single-cell microfluidics experiments to confirm some of the metabolite time traces using fluorescent reporters.

4) The metabolite flux analysis you describe is very hard to follow, mostly because the details of the methodology are in another paper. That paper is listed as 'in press', and could not be found online. This manuscript or a citation to the published article should be provided along with the revised version of your paper and if it is not already published by then, we urge you to post the article on bioRχiv.

[Editors' note: further revisions were requested prior to acceptance, as described below.]

Thank you for submitting your article *Saccharomyces cerevisiae* goes through distinct metabolic phases during its replicative lifespan" for consideration by *eLife*. Your article has been reviewed by two peer reviewers, and the evaluation has been overseen by a Reviewing Editor and Naama Barkai as the Senior Editor. The reviewers have opted to remain anonymous.

The reviewers have discussed the reviews with one another and the Reviewing Editor has drafted this decision to help you prepare a revised submission.

Summary:

Both reviewers agree that the manuscript has improved during the revision and that most of the issues have been adequately addressed by the authors. However, there are two remaining issues that need attention before the paper can be published in *eLife*.

1) The new Figure 2—figure supplement 8 needs further clarification: "293 cells (253 washed).…" what does "washed" mean? Why do the authors only report "median" lifespan here? As most lifespan studies report the "mean" rather than the "median", it is important that the mean lifespans is reported as well. Why are there error bars (shaded area) for lifespan curves? Furthermore, a description of the microfluidic experiments needs to be added in the Materials and methods section.

2) The authors hypothesize that the increase in cellular volume with age is the cause of metabolic changes. The reviewers understand the authors' claim that a larger cell has a lower surface-to-volume ratio (and the effect is not through a simple dilution). But the reviewers questioned whether the changes in the surface-to-volume ratio can be the cause of the metabolic changes since then it should occur before or at least at the same time with the metabolic changes. In Figure 2—figure supplement 1, the volume change is very small in the first 10 hours. Does this small volume change lead to a significant enough change of the surface-to-volume ratio to explain a 50% decrease of glucose uptake that occurs in 10 hours? Further discussions and clarifications will be needed.

---

## [Author Response]

Essential revisions:1) The paper is largely descriptive, without getting too much into causality. Some simple tests should be performed to either refute or support some of the authors' hypotheses. For example, to address the role of glycerol, what is the replicative lifespan of mutants that are impaired in glycerol synthesis (e.g., gpd1 mutants)?

Note that this paper (as “Research Advance”) was mainly meant to complement our earlier work, where we had presented proteome and transcriptome data during replicative aging, by metabolic profiling data.

Seeking further support for our findings, as suggested, we have looked in the role of glycerol. We had inferred an increased glycerol production as cells age. A deletion of GPD1 (glycerol 3-phosphate dehydrogenase, which is the rate limiting step in the synthesis of glycerol) should thus likely reduce the replicative lifespan, which is indeed the case (i.e. the lifespan reduces from 28 to around 12 generations, PMID: 12391171 Figure 7A). We now mention this in the manuscript.

Second, we had inferred that aged cells take up ethanol, likely produced by the daughter cells that are simultaneously present in the experiments. Seeking independent evidence for this, we performed microfluidic experiments, where we determined the lifespan of cells grown solely in glucose medium and compared their lifespan with the lifespan of cells grown in glucose medium supplemented with ethanol. Here, we found that the ethanol supplementation increased yeast’s lifespan (from 23 to 28 generations), which could serve as an indication that aged cells might indeed take up ethanol. We added this new data as new Figure 2—figure supplement 8 and refer to these data in the main text.

2) The authors use a series of complicated mathematical models to indirectly infer the metabolite level, physiological parameters and intracellular metabolic fluxes at different ages from the raw data. Given that quite a number of assumptions and approximations have been used in their models, it is hard to fully trust the inferred results without other supporting evidence. Additional controls and further validation will be needed: (a) Unlabeled cells cultured in column with non-fermentable carbon source would serve as a good control for the respiratory state and can be used to reflect the significance of the changes during aging. (b) Use another population-level aging approach, such as Mother-Enrichment Program, to reproduce some of the metabolite data. It will not be necessary to reproduce all the data, but a couple of important metabolites at a few time points would be important.

As requested by the reviewers, for this revision, we have performed further experiments and analyses. Note in this context also the additional evidence presented in response to comment 1.

The reviewer suggested to confirm the shift towards a respiratory state by using non-fermentable carbon sources. Along this line, we compared, using principle component analysis, our previously generated proteome data of aging cells with proteome data of two unlabeled *S. cerevisiae* strains (one having a fermentative and one a respiratory phenotype). The fermentative proteome was obtained from a culture of *S. cerevisiae* wildtype and the respiratory proteome from a *S. cerevisiae* strain that lacks all usual hexose transporters but has only a chimeric version of the hexose transporters Hxt1 and Hxt7. The later strain is otherwise isogenic to the wildtype strain but in high-glucose conditions has a low glucose uptake rate and thus respiratory phenotype (cf. PMID: 15345416). In line with our finding of a shift towards a more respiratory metabolism, this analysis showed that also the proteome of aging cells shifts towards a more respiratory phenotype. We added the results of this new analysis as Figure 2—figure supplement 7.

Second, the reviewer suggested the use of the mother-enrichment program (MEP) to reproduce some of the metabolite data. We evaluated the feasibility of the MEP for generating metabolome data. Therefore, we also consulted with Daniele Novarina from the group of Michael Chang who has worked for several years with the MEP. On the basis of these discussions, we had to conclude that the MEP would not be suitable to generate metabolite data of aged cells. The reasons for this are the following:

1) The MEP suffers from the fact that “escapers” can occur, i.e. mutations, through which daughter cells can divide as well. The escaper rate is in the order of 10^-6^. For this reason, researchers typically only work with very low cell numbers. However, for the metabolome analyses we need in the order of 10^7^ cells. Thus, when using the MEP with such high numbers of cells, over the course of the aging experiment, the culture would be outgrown by daughter cells.

2) If one aims to generate a high amount of cells (as required for metabolome analyses) with a high replicative age (i.e. 64 h), these cells would relatively fast deplete the glucose in the medium during cultivation (i.e. aging). This would require frequent sub-culturing of the cells which has been shown to introduce stress (PMID 17464066) and could thus introduce artifacts.

Thus, we feel that the MEP program would not be capable to generate the large amounts of aged cells, as required for metabolome analyses, in an artifact-free manner.

As an alternative, we explored single-cell metabolite sensors in combination with microfluidics to assess metabolite levels by alternative means. Previously, we had used an ATP FRET sensor to study ATP dynamics during the yeast’s cell cycle (PMID: 27989441). This sensor’s signal, however, is not only dependent on ATP, but also on pH (PMID: 19720993). It turns out that the intracellular pH drops during aging (PMID: 23172144), exactly in the pH range where the ATP FRET signal is affected. Thus, measurements with this sensor will unfortunately not report the correct ATP values during aging.

Still trying to challenge and validate our results, we did the following: We had earlier found that the concentration of the metabolite fructose-1,6-bisphospate (FBP) strictly correlates with the metabolic flux through the glycolytic pathway (PMID: 22129078). This robust correlation was found to hold also during sudden dynamic flux changes (PMID: 16672504) and across different yeast strains (PMID: 21205161). If the metabolite levels and metabolic fluxes, inferred for the aging cells, were consistent, then we could expect that the inferred FBP levels and glycolytic fluxes should comply with this correlation. Indeed, if we plot the inferred FBA concentrations as a function of glycolytic flux, then these data points exactly fall onto the earlier established correlation. In this context it is important to highlight that the inference of the metabolite concentrations and physiological rates are completely independent from each other. Thus, this observation serves as good indication for the validity of both data sets. We added a new Figure 2—figure supplement 6 showing the results of this analysis.

In summary, we provide the following independent supporting evidence for our inferred findings:

**-** Metabolite concentrations: The mathematically inferred concentrations for (young) daughter cells match the ones directly determined from a culture of (young) labeled cells.

- Metabolite concentrations/flux: The independently inferred FBP concentrations and glycolytic fluxes of aged cells fall onto an earlier established correlation.

- Physiological rates: The mathematically inferred physiological rates for (young) daughter cells match the ones directly determined from cultures of (young) labeled and unlabeled cells.

- Growth rate: The inferred decrease in growth rate with cell age is qualitatively in line with the decrease in budding rate determined in a microfluidic device.

- Shift towards respiration: The inferred shift from a respiratory towards a fermentative metabolism was shown to also occur on the proteome level.

- Production of glycerol: A deletion of GPD1 reduces yeast’s lifespan.

- Uptake of ethanol: Supplementing ethanol to the medium increases yeast’s lifespan.

3) In Figure 2 and Figure 2—figure supplement 2C, all the metabolite levels drop sharply within 0 – 10 hour (the first time point collected from the column). This is surprisingly early in the aging process. In addition, even some of the physiological parameters, such as the growth rate, drop drastically at 10-hour time point (Figure 2B). This is strikingly different from the time trace they obtained using microfluidics, where there is no change in the first 20 hours (Figure 2—figure supplement 5). This discrepancy raises the suspicion that these metabolic changes may be simply a part of the stress response induced by the column-based culturing condition. Additional controls would be needed to confirm that the column-based culturing condition is not stressful, e.g. using stress responsive genes that will not be induced during the first 10 hours of aging in microfluidics. Ideally, one would hope to see single-cell microfluidics experiments to confirm some of the metabolite time traces using fluorescent reporters.

The reviewer is surprised by the changes that occur already at a relatively young age. Please note that with several other experimental methods early-age changes were also reported. For instance, see the figure from PMID: 28357364, a paper in which age-related changes during budding yeast replicative aging were reviewed. Here, many phenotypic changes occur already very early in life (in the figure the 90-100% population viability corresponds to roughly the first 10 h of yeast’s replicative lifespan). Thus, it is not so unexpected to see changes at early time points. We now mention this in the main text.

The reviewer also raised the question whether the cells would be stressed. In our previous *eLife* paper, we indeed obtained indication that the biotin-labeling of cells introduces stress. However, the proteome and transcriptome data had indicated that this stress is gone 5-8 hours after loading the cells onto the column (note, the earliest time point we report here is 10 h). Further, as we had shown in our previous paper by means of a comparison of lifespan curves, the cells on the column age *normally* (cf. Figure 1B in previous paper). Last, our earlier generated proteome data from the column-based experiments were consistent with protein expression patterns acquired with other methods to generate aged-cells. Thus, we do not expect that stress, introduced by the experimental procedure, is generating artifacts in our data.

Second, the reviewer pointed to the discrepancy between the ‘growth rates’ of aged cells determined from the column-based experiments and the ones from the microfluidics experiment. Please note that the “growth rate” of cells from the column was determined on the basis of the increase in measured *cell dry mass*, while in the microfluidics we assessed the time between the appearances of two buds (daughter cells). Although both ‘growth measures’ decrease with time, these two measures are not the same because of yeast’s asymmetric division (with bud cells having smaller volume than mother cells). Thus, the budding rate cannot be directly converted into the cell-dry-weight-determined growth rate. We now make this clearer in the revised version of the manuscript.

Lastly, the reviewer commented on the drop of metabolite levels within the first 10 h. Triggered by this comment, we reevaluated our deconvolution analysis. Earlier, we had assumed a volume for daughter cells of 25 fL (which corresponds to the volume of daughter cells at the time they detach from their mother). However, an analysis of the acquired flow cytometric forward scatter data and a comparison between daughter and aged cells (of known volume) showed that at the time when the metabolomics samples were taken the daughter cell volume rather corresponds to 40 fL (i.e. the initial volume of mother cells). Thus, we re-did the linear regression using a volume of 40 fL for daughter cells. The inferred metabolite concentrations for daughters are (now even more) similar to concentrations obtained from a batch culture of labeled cells before they were loaded on the column (Figure 2—figure supplement 2B). Note that in an effort to increase consistency between Figure 2A and B and Figure 3, we now use the concentrations determined from the aforementioned batch culture of labeled cells as time point 0. We have updated the figures (Figure 2A and Figure 2—figure supplement 2) respectively.

4) The metabolite flux analysis you describe is very hard to follow, mostly because the details of the methodology are in another paper. That paper is listed as 'in press', and could not be found online. This manuscript or a citation to the published article should be provided along with the revised version of your paper and if it is not already published by then, we urge you to post the article on bioRxv.

The manuscript is now published in Nature Metabolism (https://doi.org/10.1038/s42255-018-0006-7). We included its full bibliography. This will further help the reader to better grasp the methods that we applied here.

[Editors' note: further revisions were requested prior to acceptance, as described below.]Summary:Both reviewers agree that the manuscript has improved during the revision and that most of the issues have been adequately addressed by the authors. However, there are two remaining issues that need attention before the paper can be published in eLife.1) The new Figure 2—figure supplement 8 needs further clarification: "293 cells (253 washed).…" what does "washed" mean? Why do the authors only report "median" lifespan here? As most lifespan studies report the "mean" rather than the "median", it is important that the mean lifespans is reported as well. Why are there error bars (shaded area) for lifespan curves? Furthermore, a description of the microfluidic experiments needs to be added in the Materials and methods section.

In the caption of the Figure 2—figure supplement 8 (now Figure 2—figure supplement 12), we clarified what the word “washed” used in the figure means. It means the number of cells washed out from the microfluidic device during the experiment. Further, in the previous version of the manuscript, we had reported only the median lifespans. As requested by the reviewers, in this revision we now also report the mean lifespans and added these numbers to the caption. In essence, the conclusion remains unaltered. In the caption, we now also explain the meaning of the shaded areas accompanying the survival curves. In the Materials and methods section, we created a separate paragraph explaining the replicative lifespan experiment and the analysis.

2) The authors hypothesize that the increase in cellular volume with age is the cause of metabolic changes. The reviewers understand the authors' claim that a larger cell has a lower surface-to-volume ratio (and the effect is not through a simple dilution). But the reviewers questioned whether the changes in the surface-to-volume ratio can be the cause of the metabolic changes since then it should occur before or at least at the same time with the metabolic changes. In Figure 2—figure supplement 1, the volume change is very small in the first 10 hours. Does this small volume change lead to a significant enough change of the surface-to-volume ratio to explain a 50% decrease of glucose uptake that occurs in 10 hours? Further discussions and clarifications will be needed.

While the surface-to-volume ratio changes with increased cell size, the cell size increase is indeed too small to fully account for the decrease in glucose uptake rate. Thus, additional factors are required to explain the observation of the decreased glucose uptake rate. We thus changed the respective sentence in the Discussion. It now reads:

“The increase We hypothesize that the increase in cellular volume (and the accompanying decrease in surface area per cell volume) with age (cf. Figure 2—figure supplement 1) could be in part is responsible for the observed decrease in the volumetric (i.e. dry weight specific) substrate influx, next to possibly altered transporter expression with age (Kamei et al., 2014)”.

We thank the reviewers for pointing this out.